# RIFLEx: A Free Lunch for Length Extrapolation in Video Diffusion Transformers

**Min Zhao** [1]  **Guande He** [2]  **Yixiao Chen** [1 3]  **Hongzhou Zhu** [1 3]  **Chongxuan Li** [4 5 6]  **Jun Zhu** [1 3 7]

## Abstract

Recent advancements in video generation have enabled models to synthesize high-quality, minute-long videos. However, generating even longer videos with temporal coherence remains a major challenge and existing length extrapolation methods lead to *temporal repetition* or *motion deceleration*. In this work, we systematically analyze the role of frequency components in positional embeddings and identify an *intrinsic frequency* that primarily governs extrapolation behavior. Based on this insight, we propose **RIFLEx**, a *minimal yet effective* approach that reduces the intrinsic frequency to suppress repetition while preserving motion consistency, without requiring any additional modifications. RIFLEx offers a true *free lunch*—achieving high-quality $2\times$ extrapolation on state-of-the-art video diffusion transformers in a completely *training-free* manner. Moreover, it enhances quality and enables $3\times$ extrapolation by minimal fine-tuning without long videos. Project page and codes: https://riflex-video.github.io/.

## 1. Introduction

Recent advances in video generation (Brooks et al., 2024; Bao et al., 2024; Yang et al., 2024; Kong et al., 2024; Zhao et al., 2023; Team, 2024b; Zhao et al., 2024) enable models to synthesize minute-long video sequences with high fidelity and coherence. A key factor behind this progress is the emergence of diffusion transformers (Peebles & Xie,

2023; Bao et al., 2023), which combines the scalability of diffusion models (Sohl-Dickstein et al., 2015; Ho et al., 2020; Song et al., 2021) with the expressive power of transformers (Vaswani, 2017).

Despite these advancements, generating longer videos with high-quality and temporal coherence remains a fundamental challenge. Due to computational constraints and the sheer scale of training data, existing models are typically trained with a fixed maximum sequence length, limiting their ability to extend content. Consequently, there is increasing interest in length extrapolation techniques that enable models to *generate new and temporally coherent content that evolves smoothly over time without training on longer videos*.

However, existing extrapolation strategies (Chen et al., 2023b; bloc97, 2023; Peng et al., 2023; Lu et al., 2024b; Zhuo et al., 2024), originally developed for text and image generation, fail when applied to video length extrapolation. Our experiments show that these methods exhibit distinct failure patterns: *temporal repetition* and *slow motion*. These limitations suggest a fundamental gap in understanding how positional encodings influence video extrapolation.

To address this, we isolate individual frequency components by zeroing out others and fine-tuning the target video model. We find that high frequencies capture short-term dependencies and induce temporal repetition, while low frequencies encode long-term dependencies but lead to motion deceleration. Surprisingly, we identify a consistent *intrinsic frequency component* across different videos from the same model, which primarily dictates repetition patterns among all components during extrapolation.

Building on this insight, we propose **R**educing **I**ntrinsic **F**requency for **L**ength **Ex**trapolation (**RIFLEx**), a *minimal yet effective* solution that lowers the intrinsic frequency to ensure it remains within a single cycle after extrapolation. Without any other modification, it suppresses temporal repetition while preserving motion consistency. As a byproduct, RIFLEx provides a principled explanation for the failure modes of existing approaches and offers insights that naturally extend to spatial extrapolation of images.

Extensive experiments on state-of-the-art video diffusion transformers, including CogVideoX-5B (Yang et al., 2024)

---

[1]Dept. of Comp. Sci. & Tech., BNRist Center, THU-Bosch ML Center, Tsinghua University. [2]The University of Texas at Austin. [3]ShengShu. [4]Gaoling School of Artificial Intelligence Renmin University of China Beijing, China. [5]Beijing Key Laboratory of Research on Large Models and Intelligent Governance. [6]Engineering Research Center of Next-Generation Intelligent Search and Recommendation, MOE. [7]Pazhou Laboratory (Huangpu). Correspondence to: Chongxuan Li <chongxuanli@ruc.edu.cn>, Jun Zhu <dcszj@tsinghua.edu.cn>.

*Proceedings of the $42^{nd}$ International Conference on Machine Learning*, Vancouver, Canada. PMLR 267, 2025. Copyright 2025 by the author(s).

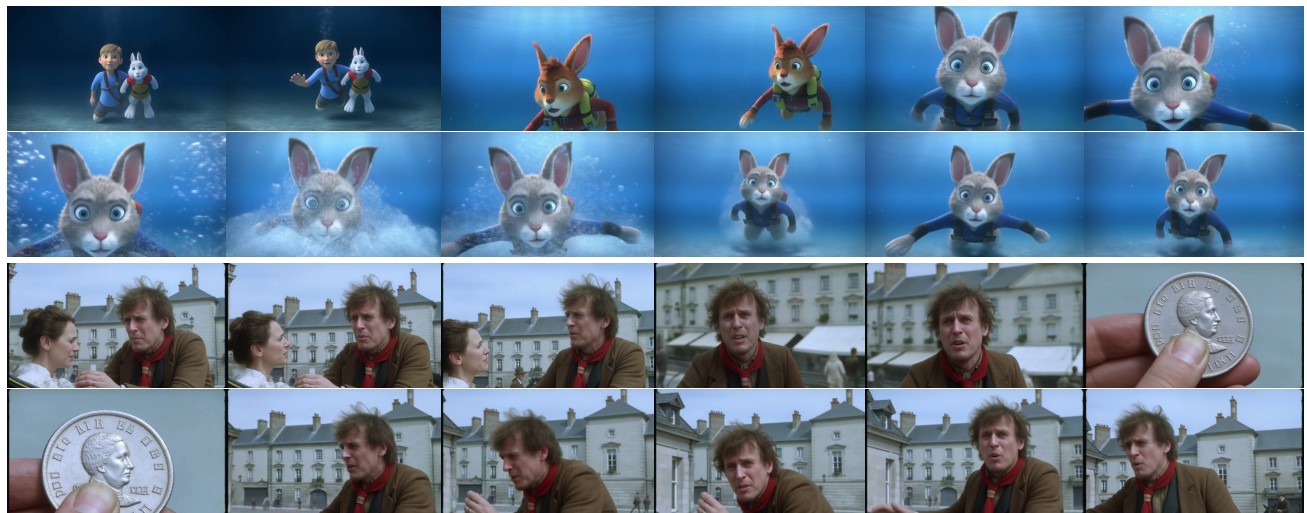

(a) $2\times$ temporal extrapolation from 129 to 261 frames.

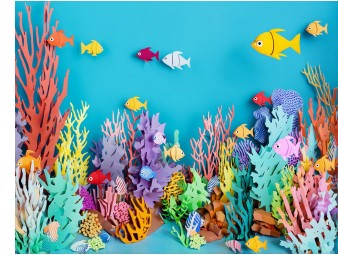
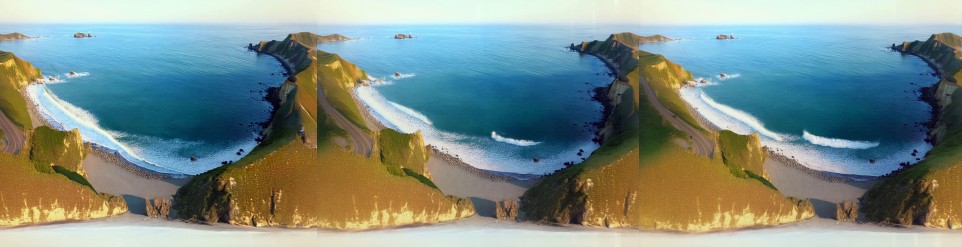

(b) From $480\times720$ to $960\times1440$.      (c) $2\times$ temporal and spatial extrapolation from $480 \times 720 \times 49$ to $960 \times 1440 \times 97$.

*Figure 1.* **Visualization of RIFLEx for** $2\times$ **temporal, spatial, and combined extrapolation.** Our base models are (a) Hunyuan-Video (Kong et al., 2024) and (b-c) CogVideoX-5B (Yang et al., 2024), where we do not use any videos longer or larger than those used for pre-training. More demos and all the prompts used in this paper are listed in Appendix B and the supplementary materials, respectively.

and HunyuanVideo (Kong et al., 2024), validate the effectiveness of RIFLEx (see Fig. 1). Remarkably, for $2\times$ extrapolation, RIFLEx enables high-quality and natural video generation in a completely *training-free* manner. When fine-tuning is applied with only 20,000 original-length videos—requiring just 1/50,000 of the pre-training computation—sample quality is further improved, and the effectiveness of RIFLEx extends to $3\times$ extrapolation. Moreover, RIFLEx can also be applied in the spatial domain simultaneously to extend both video duration and spatial resolution.

Our key contributions are summarized as follows:

- We provide a *comprehensive understanding* of video length extrapolation by analyzing the failure modes of existing methods and revealing the role of individual frequency components in positional embeddings.

- We propose **RIFLEx**, a *minimal yet effective* solution that mitigates repetition by properly reducing the intrinsic frequency, without any additional modifications.

- RIFLEx offers a true *free lunch*—achieving high-quality $2\times$ extrapolation on state-of-the-art video diffu-

sion transformers in a completely *training-free* manner. Moreover, it enhances quality and enables $3\times$ extrapolation by minimal fine-tuning without long videos.

## 2. Background

### 2.1. Video Generation with Diffusion Transformers

Given a data distribution $p_{\text{data}}$, diffusion models (Sohl-Dickstein et al., 2015; Ho et al., 2020; Song et al., 2021) progressively perturb the clean data $\boldsymbol{x}_0 \sim p_{\text{data}}$ with a transition kernel $q_{t|0}(\boldsymbol{x}_t|\boldsymbol{x}_0) = \mathcal{N}(\alpha_t \boldsymbol{x}_0, \sigma_t^2 \boldsymbol{I})$, i.e., $\boldsymbol{x}_t = \alpha_t \boldsymbol{x}_0 + \sigma_t \boldsymbol{\epsilon}$, where $t \in [0, T]$, $\alpha_t, \sigma_t$ are pre-defined noise schedule, and $\boldsymbol{\epsilon} \sim \mathcal{N}(\boldsymbol{0}, \boldsymbol{I})$ is Gaussian noise. Under proper designs of $\alpha_t, \sigma_t$, the distribution of $\boldsymbol{x}_T$ is tractable, e.g., a standard Gaussian.

A generative model is obtained by reversing this process from $t = T$ to 0, whose dynamic is characterized by the score function $\nabla_{\boldsymbol{x}_t} \log q_t(\boldsymbol{x}_t)$. The score function is usually parameterized by a neural network $\boldsymbol{s}_{\boldsymbol{\theta}}(\boldsymbol{x}_t, t)$ and learned with the denoising score matching (Vincent, 2011): $\mathbb{E}_{t, \boldsymbol{x}_0, \boldsymbol{\epsilon}}[w(t)\|\boldsymbol{s}_{\boldsymbol{\theta}}(\boldsymbol{x}_t, t) - \nabla_{\boldsymbol{x}_t} \log q_{t|0}(\boldsymbol{x}_t|\boldsymbol{x}_0)\|^2]$,

where $w(t)$ is a weighting function. The de facto approach for modeling video data via diffusion models is to first encode the video data into sequences of latent space, then perform diffusion modeling with transformer-based neural network (Peebles & Xie, 2023; Bao et al., 2023).

## 2.2. Position Embedding in Diffusion Transformers

A position embedding is a fixed or learnable vector-valued function $\boldsymbol{f}$ that maps a $n$-axes position vector $\boldsymbol{p} \in \mathbb{N}_+^n$ to some representation space. This position information can be incorporated into transformers through various mechanisms, such as through additive (Vaswani, 2017; Raffel et al., 2020; Press et al., 2021) or multiplicative (Su et al., 2021) operations with other input or hidden embeddings.

**Rotary Position Embedding (RoPE)** (Su et al., 2021) has emerged as the predominant method in transformers. RoPE encodes relative positional information by interacting with two absolute position embeddings within the attention mechanism. Specifically, for a sequence indexed by a single axis (i.e. $n = 1$), given an input $\boldsymbol{x} \in \mathbb{R}^d$ with position $p \in \mathbb{N}_+$, RoPE maps it to an absolute-position encoded embedding on $\mathbb{R}^{d'}$ with $d' \leq d$, i.e.,

$$\boldsymbol{f}^{\mathrm{RoPE}}(\boldsymbol{x}, p, \boldsymbol{\theta})_j = \begin{bmatrix} \cos p\theta_j & -\sin p\theta_j \\ \sin p\theta_j & \cos p\theta_j \end{bmatrix} \begin{bmatrix} x_{2j} \\ x_{2j+1} \end{bmatrix}. \quad (1)$$

where $\boldsymbol{\theta} \in \mathbb{R}^{d'/2}$ with $\theta_j = b^{-2(j-1)/d'}$ for $j = 1, \ldots, d'/2$. Here, $\boldsymbol{\theta}$ represents the frequencies for all dimensions of the RoPE embedding and $b$ is a hyperparameter that adjusts the base frequency. It can be verified that the dot product between two RoPE-embedded vectors encodes the relative positional information between them. In practice, RoPE is applied to the query and key vectors before the dot product operation in the attention mechanism, and thus the result attention matrix encodes the relative positional information.

**RoPE with Multiple Axes.** RoPE can be extended to multi-axes position vector $\boldsymbol{p} \in \mathbb{N}_+^n$ for $n > 1$. One popular practice is to encode each axis independently. For example, consider a video input $\boldsymbol{x} \in \mathbb{R}^d$ with three-dimensional coordinates $(t, h, w)$, there are three axis-specific parameters $\boldsymbol{\theta}^t, \boldsymbol{\theta}^h, \boldsymbol{\theta}^w$. Single-axis RoPE, as defined in Eqn. (1), is then applied separately along the feature dimension with these three parameters. The final multi-axes RoPE is obtained by concatenating the three single-axis RoPE embeddings.

## 2.3. Length Extrapolation with RoPE

In this section, we briefly recap the techniques for length extrapolation with RoPE adopted in text and image.

The most straightforward approach, **Position Extrapolation (PE)**, extends the input sequence length without modifying the positional encoding, which purely relies on the

generalization ability of the network and the positional encoding. Whereas **Position Interpolation (PI)** (Chen et al., 2023b) uniformly down-scales all frequencies in RoPE embedding to match the training sequence length. In specific, the new RoPE frequencies are calculated as $\boldsymbol{\theta}^{\mathrm{PI}} = \boldsymbol{\theta}/s$, where $s = L'/L$, and $L, L'$ is the sequence length for training and inference, respectively.

A key limitation of both PE and PI is their reliance on training at the target sequence length, otherwise, the performance degrades drastically. To address this, **NTK-Aware Scaled RoPE (NTK)** (bloc97, 2023) combines the ideas of both position extrapolation and interpolation. Specifically, NTK adjusts the base frequency $b$ for all dimensions as:

$$\theta_j^{\mathrm{NTK}} = (\lambda b)^{-2(j-1)/d'}, \lambda = s^{d'/(d'-2)}, j = 1, \ldots, d'/2, \quad (2)$$

where $s = L'/L$. NTK effectively applies PE for high frequencies and PI for low frequencies, enabling training-free extrapolation. (bloc97, 2023; Zhuo et al., 2024).

**YaRN** (Peng et al., 2023) introduces a fine-grained base frequency adjustment strategy. It first categorizes all frequencies into three groups based on *the number of cycles elapsed over the training length*, defined as $r_j = (2\pi)^{-1} L\theta_j$. Given two pre-determined thresholds $\alpha, \beta$ with $r_{d'/2} \leq \alpha < \beta \leq r_1$, YaRN adjusts the RoPE frequencies as:

$$\theta_j^{\mathrm{YaRN}} = \gamma(r_j)\theta_j + (1 - \gamma(r_j))\frac{\theta_j}{s}, \ j = 1, \ldots, d'/2,$$

$$\gamma(r_j) = \begin{cases} 1, & \text{if } r_j > \beta, \\ 0, & \text{if } r_j < \alpha, \\ \frac{r_j - \alpha}{\beta - \alpha}, & \text{otherwise,} \end{cases} \quad (3)$$

In practice, YaRN exhibits better training-free extrapolation performance compared to NTK and can achieve great performance with a relatively small fine-tuning budget on target length (Peng et al., 2023).

**Length Extrapolation in Image Diffusion Transformers.** Image diffusion transformers have two key characteristics related to RoPE: (1) image data is represented as a sequence with height and width axes, and (2) an iterative diffusion sampling procedure. These characteristics inform specific length extrapolation techniques for image diffusion models.

For multi-axes sequence, RoPE is independently applied to each axis, allowing length extrapolation techniques like NTK and YaRN to be used separately on height and width, termed Vision NTK and Vision YaRN (Lu et al., 2024b). For sampling, different RoPE adjustments can be employed at various diffusion timesteps. For instance, **Time-aware Scaled RoPE (TASR)** (Zhuo et al., 2024) leverages PI at large timesteps to preserve global structure while using NTK at smaller timesteps to enhance visual quality.

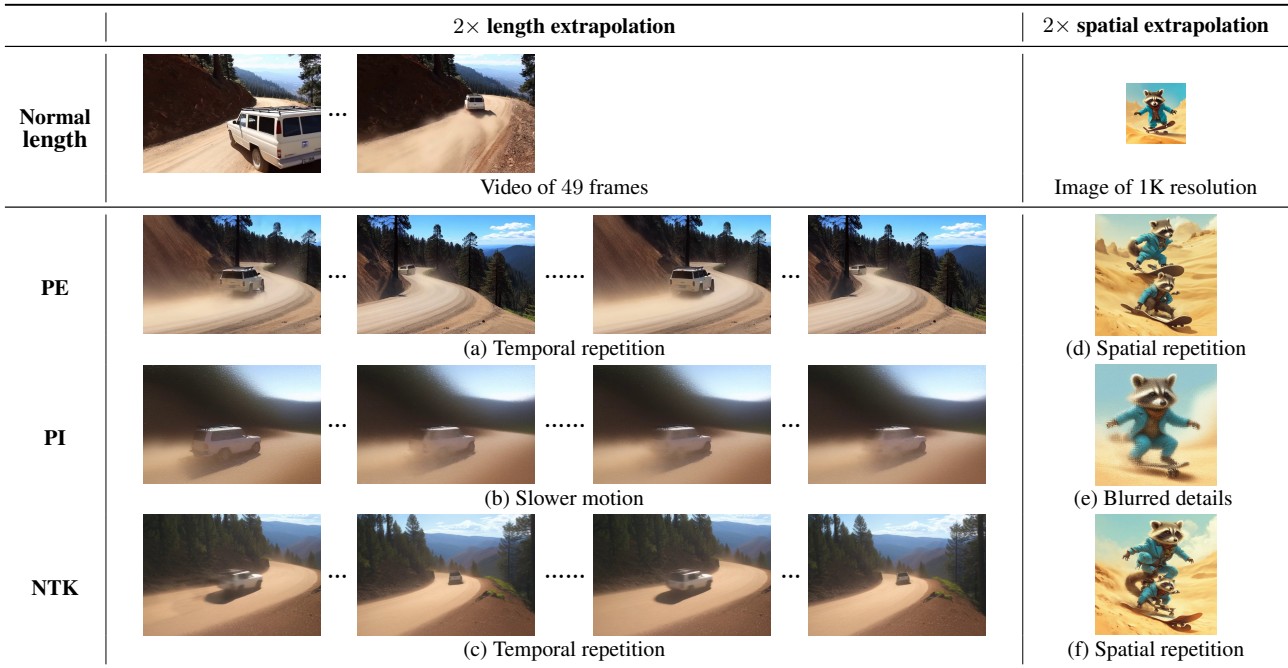

*Figure 2.* **Visualization of existing methods for 2× extrapolation in video and image generation.** The base models CogVideoX-5B (Yang et al., 2024) and Lumina-Next (Zhuo et al., 2024) are trained to sample videos of up to 49 frames and images of up to 1K resolution, respectively. Existing methods lead to *temporal repetition* or *slower motion* in video extrapolation and *spatial repetition* or *blurred content* in image extrapolation, respectively. Please refer to Appendix C for more results and details.

## 3. Method

Our goal is to understand and solve the video length extrapolation problem thoroughly. We first highlight the intriguing failure patterns of existing methods, analyze the role of different frequency components in positional embeddings, and identify an *intrinsic frequency*. Based on this, we derive a minimal solution that enables length extrapolation: *reducing the intrinsic frequency*. As byproducts, our method not only provides a principled explanation for the failure of existing approaches in video extrapolation but also offers insights applicable to spatial extrapolation in images.

### 3.1. Failure Patterns of Existing Methods

Although the term "extrapolation" is widely used across different domains, its role in video generation is fundamentally different from text and images. In video generation, the objective is to create *new and temporally coherent content that evolves smoothly over time*. In contrast, text extrapolation primarily extends the context window, while image extrapolation typically involves adding high-resolution details rather than generating meaningful new content.

As a result, extrapolation strategies developed for text and images fail in video length extrapolation, exhibiting intriguing failure patterns, as illustrated in Fig. 2. To better understand these patterns, we also conduct the counterparts on image spatial extrapolation, revealing parallels to video.

PE, which directly extends positional encoding beyond the training range, leads to *temporal repetition*, causing videos to loop instead of progressing naturally (Fig. 2a). A similar phenomenon occurs in image generation, where *spatial repetition* occurs instead of generating new content.

Conversely, PI (Chen et al., 2023b), which compresses positional encodings within the training range, leads to *slow motion* by stretching frames over time (Fig. 2b). While this approach preserves structural coherence, it lacks temporal novelty. In image generation, this results in *blurred details* rather than new content (Fig. 2e).

As shown in Fig.2c, NTK (bloc97, 2023) also induces *temporal repetition*, failing to generate meaningful motion progression. In image generation, it causes *spatial repetition* (Fig. 2f). While other methods (Peng et al., 2023; Lu et al., 2024b; Zhuo et al., 2024) differ from NTK in implementation, they invariably suffer from one or both of these two failure patterns: either motion deceleration or content repetition (see Appendix C for further analysis).

Beyond revealing these limitations, our findings provide an intuitive understanding of how positional embeddings fundamentally shape temporal motion, motivating our in-depth frequency component analysis in the next section.

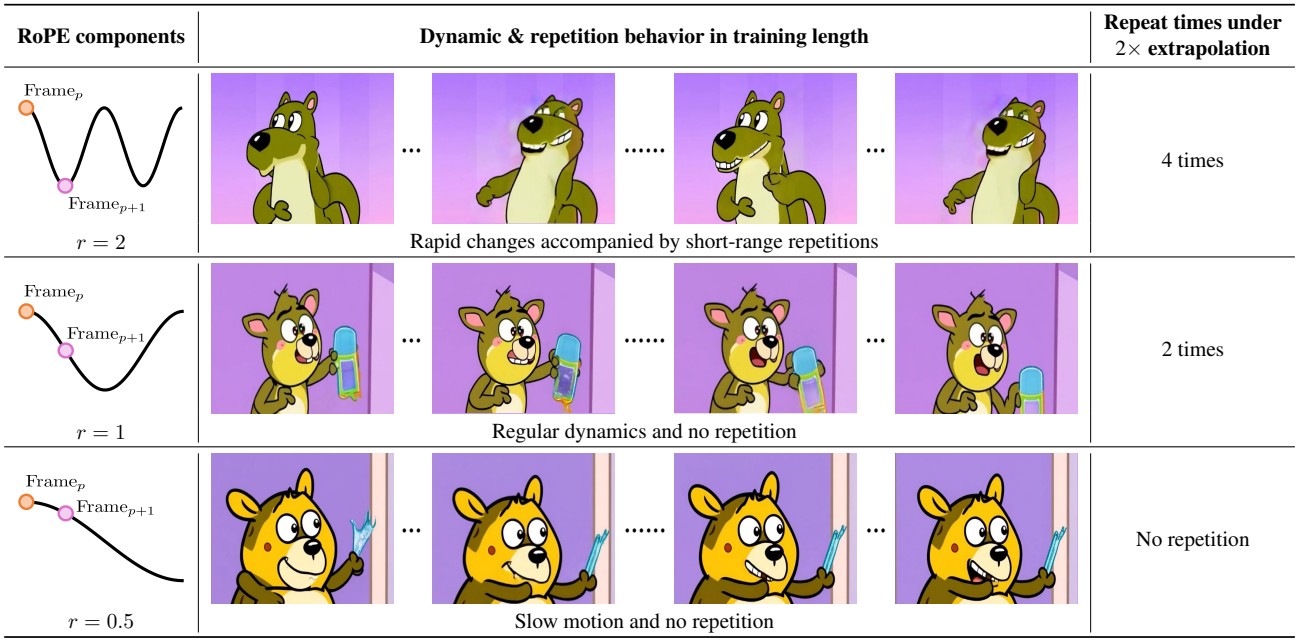

| RoPE components | Dynamic & repetition behavior in training length | Repeat times under $2\times$ extrapolation |
|---|---|---|
| $\text{Frame}_p$ $\text{Frame}_{p+1}$ $r = 2$ | Rapid changes accompanied by short-range repetitions | 4 times |
| $\text{Frame}_p$ $\text{Frame}_{p+1}$ $r = 1$ | Regular dynamics and no repetition | 2 times |
| $\text{Frame}_p$ $\text{Frame}_{p+1}$ $r = 0.5$ | Slow motion and no repetition | No repetition |

*Figure 3.* **Visualization of frequency components and their roles in video generation.** High frequencies capture rapid movements and short-term dependencies, inducing temporal repetition, while low frequencies encode long-term dependencies with slow motion.

## 3.2. Frequency Component Analysis in RoPE

We begin by analyzing the role of individual frequency components in RoPE (Su et al., 2021). We follow the notation in Sec. 2.2 but focus on the time axis and omit the subscript $t$ for simplicity. We isolate specific frequency components by zeroing out others and fine-tune the target model (Yang et al., 2024) on its training length to adapt to the modified RoPE. Two key insights emerge from this analysis.

First, different frequency components $\theta_j$ capture temporal dependencies at varying scales, dictated by their *periods*:

$$N_j = \frac{2\pi}{\theta_j}, \quad (4)$$

where $j$ denotes the frequency index in RoPE. As illustrated in Fig. 3, when the frame interval exceeds $N_j$, the periodic nature of the cosine function forces positional encodings—and consequently, generated video content—to repeat. Given a training length $L$, the number of temporal repetitions can be quantified as:

$$r_j = \frac{L}{N_j} = \frac{L\theta_j}{2\pi}. \quad (5)$$

As shown in Fig. 3, when a high-frequency component has $r_j = 2$, the video completes two cycles within the training length and four cycles during $2\times$ extrapolation. In contrast, a low-frequency component with $r_j = 0.5$ remains within a single cycle even when extrapolated.

Second, frequency components influence the perceived motion speed in video generation. This effect correlates to the *rate of change* in positional encoding between consecutive (e.g., $p$-th and $(p+1)$-th) frames:

$$\Delta_j = \cos((p+1)\theta_j) - \cos(p\theta_j). \quad (6)$$

Higher frequencies (i.e., larger $\theta_j$) typically result in larger $\Delta_j$, making the model more sensitive to rapid movements. Conversely, lower-frequency components induce minimal positional encoding shifts between adjacent frames, favoring slow-motion dynamics, aligning with results in Fig. 3.

Given that each component has its own period $N_j$, a key question arises: **which frequency primarily dictates the observed repetition pattern** in length extrapolation?

We define the *intrinsic frequency component* as the one whose period $N_j$ is closest to the first observed repetition frame $N$ in a video, as it determines the overall behavior:

$$k = \arg\min_j |N_j - N|, \quad (7)$$

where $j$ denotes the frequency index in RoPE. Surprisingly, this intrinsic frequency remains consistent across different videos generated by the same model, despite slight variations in $N$. For instance, $k$ is 2 for CogVideoX-5B (Yang et al., 2024) and 4 for HunyuanVideo (Kong et al., 2024) respectively, as detailed in Appendix E.

In the rare case where a model exhibits inconsistent intrinsic frequencies across videos, we suggest treating all such

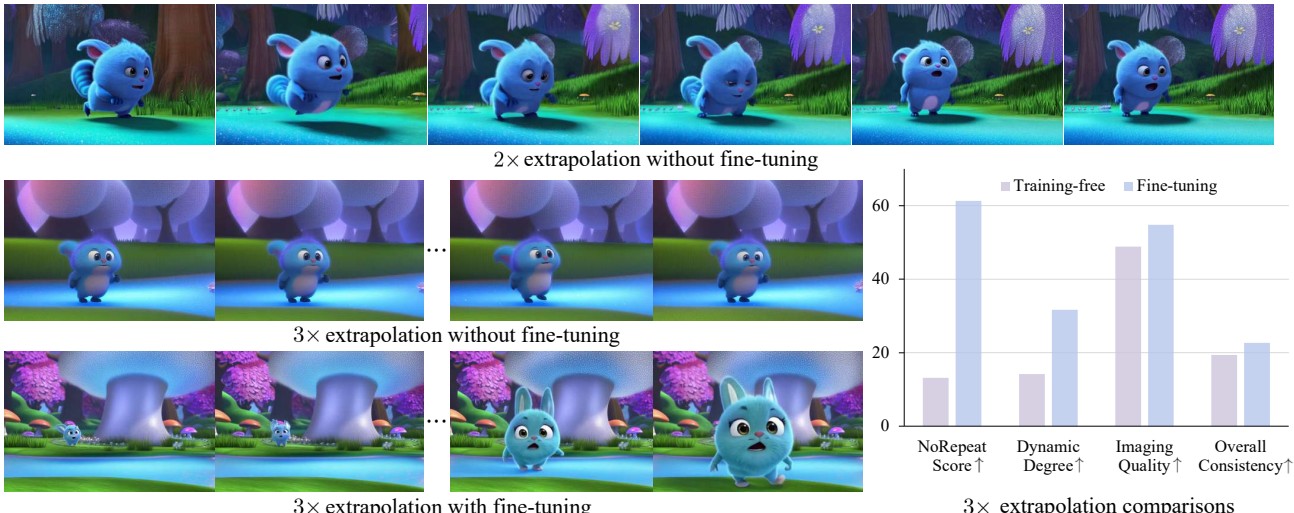

**Figure 4. Exploring the necessity of fine-tuning.** For 2× extrapolation, RIFLEx generates high-quality videos without fine-tuning. For 3× extrapolation, due to the large intrinsic frequency shift, fine-tuning is required to improve dynamic effects and visual quality.

frequencies as intrinsic. Our preliminary experiments further validate this assumption, showing that incorporating all lower-frequency components into our method maintains strong performance, as discussed in Appendix E.

### 3.3. Reducing Intrinsic Frequency: A Minimal Solution

Consider a video diffusion transformer trained on sequences of length $L$. We aim to generate videos of length $sL$ via extrapolation by a factor of $s$[1]. Based on previous findings, we propose a natural and minimal solution: **R**educing **I**ntrinsic **F**requency for **L**ength **Ex**trapolation (**RIFLEx**). RIFLEx lowers the intrinsic frequency so that it remains within a single period after extrapolation:

$$N'_k \geq Ls \Rightarrow \theta'_k \leq \frac{2\pi}{Ls}. \tag{8}$$

By setting $\theta'_k = \frac{2\pi}{Ls}$, we achieve a minimal modification. Ablation studies on other frequencies (Appendix E) confirm that modifying only the intrinsic frequency is sufficient: adjusting higher-frequency components disrupt fast motion while altering lower frequencies has negligible impact. We present RIFLEx formally in Algorithm 1.

We further investigate **whether fine-tuning is necessary** for RIFLEx. Surprisingly, for 2× extrapolation, RIFLEx can generate high-quality videos in a *training-free* manner, as shown in Fig. 4. Fine-tuning with only 20,000 original-length videos and 1/50,000 of the pre-training computation further enhances dynamic quality and visual quality.

For 3× extrapolation, the intrinsic frequency shift becomes too large, leading to a notable training-testing mismatch. This occurs because the position embeddings used during in-

---

[1]We assume $s$ is sufficiently large such that $N_k < Ls$. Otherwise, it is trivial to generate long videos by PE.

---

**Algorithm 1** RIFLEx

**Require:** The extrapolation factor $s$, frequencies $\theta_j$ in the RoPE, the first observed repetition frame $N$

1: **for** $j = 1$ to $\dfrac{d'}{2}$ **do**
2:     Compute the period of each $\theta_j$ (Eqn. (4))
3: **end for**
4: Identify the intrinsic frequency component $k$ (Eqn. (7))
5: Modify $\theta_k = \frac{2\pi}{Ls}$

---

ference deviate slightly from those seen during training due to modified frequencies. While this discrepancy does not undermine the conclusion about our non-repetition condition in Eqn. (8), it may affect visual quality since the model lacks explicit training on these specific position embeddings. Nevertheless, the fine-tuning process still succeeds, as shown in Fig. 4.

### 3.4. Principled Explanation for Existing Methods

Our findings provide a principled explanation for the failure patterns observed in Section 3.1. The repetition observed in PE and NTK (bloc97, 2023; Lu et al., 2024b) stems from their intrinsic frequency component violating the non-repetition condition in Eqn. (8). As a result, the generated video content loops instead of progressing naturally. PI (Chen et al., 2023b) and YaRN (Peng et al., 2023) cause slow motion by interpolating high-frequency components, which are crucial for fast motion. Divided by $s$ in such methods, these components cannot generate rapid movements. TASR (Zhuo et al., 2024) combines both approaches mentioned above, resulting in a mixture of temporal repetition and motion slowdown. See Appendix C for more details and experiments.

*Table 1.* **Quantitative results in length extrapolation.** The red-marked areas in the NoRepeat Score and Dynamic Degree indicate severe issues with repetition and slow motion, making other metrics meaningless. In the user study, the ratio for no extrapolation represents the proportion of users who prefer the samples of the training length over RIFLEx. The others are the corresponding ranks among all methods.

| Method | Automatic Metrics↑ | | | | User Study↓ | | |
|---|---|---|---|---|---|---|---|
| | NoRepeat Score | Dynamic Degree | Imaging Quality | Overall Consistency | Motion Quality | Visual Quality | Overall Aspects |
| CogVideoX-5B with 2× extrapolation, training-free | | | | | | | |
| No extrapolation | - | 67.5 | 64.4 | 25.8 | 66.4% | 76.0% | 70.2% |
| PE | 46.6 | 58.6 | 55.0 | 22.9 | 2.1 | 1.6 | 2.4 |
| NTK | 43.4 | 58.3 | 55.3 | 22.9 | 2.1 | 1.8 | 2.1 |
| PI | 59.0 | 5.0 | 44.3 | 19.2 | 3.7 | 4.1 | 3.8 |
| TASR | 10.8 | 26.9 | 50.5 | 21.5 | 3.3 | 3.8 | 3.6 |
| YaRN | **59.4** | 5.6 | 44.6 | 19.3 | 3.6 | 4.2 | 3.7 |
| RIFLEx (**ours**) | 54.2 | **59.4** | **56.9** | **23.5** | **1.4** | **1.5** | **1.1** |
| CogVideoX-5B with 2× extrapolation, fine-tuning | | | | | | | |
| No extrapolation | - | 65.6 | 62.7 | 25.8 | 61.8% | 66.0% | 65.0% |
| PE | 13.2 | 50.6 | 56.6 | 24.2 | 1.8 | 1.8 | 1.7 |
| RIFLEx (**ours**) | **61.3** | **54.7** | **60.4** | **25.0** | **1.2** | **1.2** | **1.3** |
| HunyuanVideo with 2× extrapolation, training-free | | | | | | | |
| No extrapolation | - | 63.0 | 65.9 | 19.6 | 62.8% | 62.0% | 61.6% |
| PE | 36.0 | **63.0** | 64.3 | **19.1** | 2.3 | 1.2 | 2.4 |
| NTK | 81.0 | 55.0 | **65.3** | 18.9 | **1.5** | 1.4 | 1.6 |
| PI | **86.0** | 11.0 | 57.4 | 18.9 | 4.3 | 2.8 | 3.8 |
| TASR | 85.0 | 18.0 | 61.3 | 19.0 | 4.2 | 2.2 | 3.4 |
| YaRN | **86.0** | 15.0 | 58.2 | 18.8 | 3.9 | 2.7 | 3.7 |
| RIFLEx (**ours**) | 72.0 | 57.0 | 65.2 | 19.0 | 1.6 | **1.1** | 1.4 |
| HunyuanVideo with 2.3× extrapolation, training-free | | | | | | | |
| NTK | 20.0 | 46.0 | **65.5** | **18.3** | 1.7 | 1.6 | 1.7 |
| RIFLEx (**ours**) | **54.0** | **51.0** | 65.0 | 18.1 | **1.3** | **1.4** | **1.3** |
| HunyuanVideo with 2× extrapolation, fine-tuning | | | | | | | |
| No extrapolation | - | 79.0 | 71.6 | 18.8 | 62.6% | 51.2% | 56.0% |
| PE | 40.0 | 74.0 | 71.6 | **18.7** | 1.9 | 1.6 | 1.8 |
| RIFLEx (**ours**) | **89.0** | **82.0** | **72.0** | 18.1 | **1.1** | **1.4** | **1.2** |

## 4. Experiments

### 4.1. Setup

We describe the dataset and evaluation setup below, with implementation details in Tab. 3 (see Appendix D).

**Datasets.** We use a private dataset of 20,000 videos for fine-tuning. For CogVideoX-5B, We adopt the VBench (Huang et al., 2024) prompts to ensure consistency with prior work (Yang et al., 2024). Due to the high computational cost of HunyuanVideo (Kong et al., 2024), we evaluate it

using 100 diverse prompts across multiple categories.

**Evaluation metrics.** Following prior work (Huang et al., 2024; Yang et al., 2024), we assess video generation using Imaging Quality, Dynamic Degree, and Subject Consistency, measuring visual quality, motion magnitude, and temporal consistency, respectively. Additionally, we introduce the NoRepeat Score, where a higher score indicates less repetition (detailed in Appendix D). We also conduct a user study with 10 participants, evaluating visual quality, motion quality, and overall preference. Motion quality reflects both repetition and slow motion. Users rank their preferences

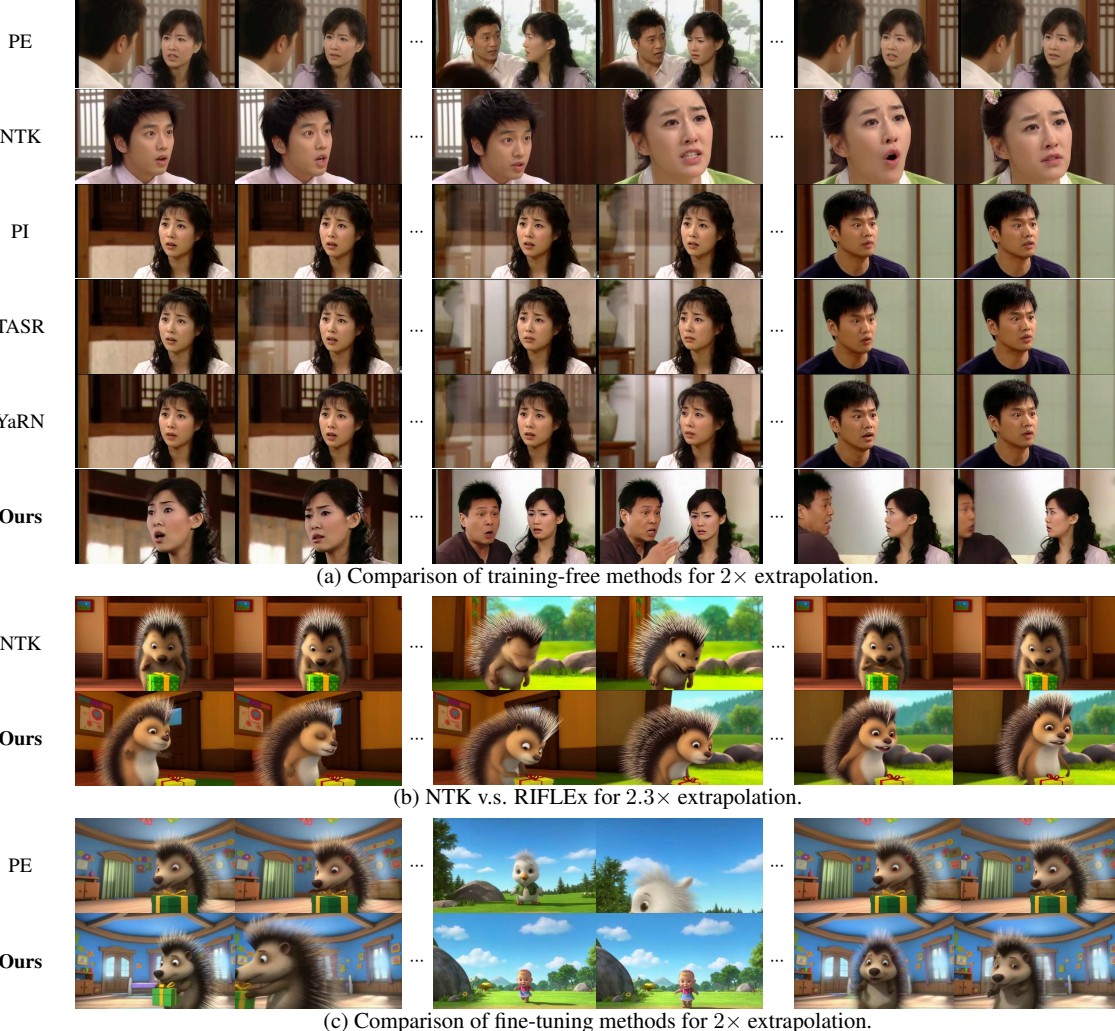

(a) Comparison of training-free methods for 2× extrapolation.

(b) NTK v.s. RIFLEx for 2.3× extrapolation.

(c) Comparison of fine-tuning methods for 2× extrapolation.

*Figure 5.* **Visualization results of length extrapolation based on HunyuanVideo.** We achieve better video quality by effectively addressing issues of slow motion and repetition. Notably, while the NTK in HunyuanVideo incidentally avoids repetition at 2× extrapolation, it still encounters significant repetition at longer extrapolations, such as 2.3× extrapolation.

among all extrapolation methods, allowing for ties. We also perform pairwise comparisons between the results of normal samples and RIFLEx. See more details in Appendix D.

### 4.2. Performance Comparison

**Results.** Quantitative results are summarized in Tab. 1. Our approach achieves superior overall performance, generating new temporal content without compromising other aspects of video quality. For example, in CogVideoX-5B, PI and YaRN suffer from slow motion, leading to lower *Dynamic Degree*, while PE and NTK experience repetition issues, resulting in lower *NoRepeat Score*. By effectively addressing both challenges, our method significantly enhances motion quality and ranks highest in user studies across all methods.

Notably, NTK coincidentally performs well for Hunyuan-Video at 2× extrapolation, but our analysis attributes this to an unintended intrinsic frequency reduction that happens to satisfy the non-repetition condition in Eqn. (8), rather than its intended mechanism. This is evident as NTK fails on CogVideo-X and HunyuanVideo with 2.3× extrapolation, reflected in its low *NoRepeat Score* in Tab. 1.

Qualitative results are shown in Fig. 5 for HunyuanVideo, with additional comparisons for CogVideoX-5B in Appendix F. Fig. 5 aligns with the quantitative findings, demonstrating our method's ability to effectively mitigate slow motion and repetition, thereby improving overall video quality.

Additionally, a minimal fine-tuning procedure requiring just 1/50,000 of the pre-training computation on short videos improves the *Dynamic Degree*, *Imaging Quality*, and *NoRepeat Score*. Finally, leveraging the strong HunyuanVideo base model, our approach achieves performance close to

that of the training length—with 56.0% and 61.6% of users preferring the training length over our method.

**Maximum extent of extrapolation.** Empirically, RIFLEx supports up to $3\times$ extrapolation, beyond which quality degrades significantly (e.g., at $4\times$ extrapolation, see Fig. 9 in Appendix). This may occur because excessive frequency reduction diminishes the effectiveness of RoPE, resulting in minimal encoding changes over the training length.

**Extension to other extrapolation types.** Video diffusion transformers typically apply 1D RoPE (see Sec. 2.2) independently to both spatial and temporal dimensions. This shared mechanism leads to analogous extrapolation challenges in both domains. Consequently, our method naturally extends to spatial extrapolation and joint temporal-spatial extrapolation, offering a unified framework for extrapolation in diffusion transformers. As shown in Fig. 1b and Fig. 1c, adjusting the intrinsic frequency for the corresponding dimensions enables resolution extrapolation and joint spatial-temporal extension. Additional demos and implementation details are provided in Appendix B and Appendix D.

## 5. Conclusion and Discussion

We provide a comprehensive understanding of video length extrapolation by analyzing the role of frequency components in RoPE. Building on these insights, we propose RIFLEx, a minimal yet effective solution that prevents repetition by reducing intrinsic frequency. RIFLEx achieves high-quality $2\times$ extrapolation on SOTA video diffusion transformers in a training-free manner and enables $3\times$ extrapolation by minimal fine-tuning without long videos.

In this paper, we primarily adopt an empirical approach—visual inspection—for intrinsic frequency identification when adapting the pre-trained video diffusion transformer. While this approach is effective for adaptation, establishing a theoretical foundation for intrinsic frequency identification is crucial. Achieving this would require fundamental research into how intrinsic frequencies emerge during the pre-training process, potentially analysis from a training-from-scratch perspective. What's more, as discussed in the main text, the $3\times$ limitation stems from diminished ability to discriminate sequential positions due to excessive frequency reduction. To further extend beyond this, it is promising to investigate the mechanism of positional encoding during training, specifically tailored for extrapolation.

## Acknowledgements

This work was supported by NSFC Projects (Nos. 62350080, 62106122, 92248303, 92370124, 92470118, 62350080, 62276149, U2341228, 62076147), Beijing NSF (No. L247030), Beijing Nova Program (No. 20230484416), Tsinghua Institute for Guo Qiang, and the High Performance Computing Center, Tsinghua University. J. Zhu was also supported by the XPlorer Prize.

## Impact Statement

This paper presents work aimed at advancing the field of video generation. It is crucial to use this technology responsibly to prevent negative social impacts, such as the creation of misleading fake videos.

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

# A. Related Work

**Length extrapolation with RoPE.** Position encoding, exemplified by the widely used RoPE, plays a crucial role in enabling length extrapolation in transformers. Prior research in both language and image domains has primarily focused on training-free methods and fine-tuning under target sequence length settings. For instance, position interpolation generally outperforms direct position extrapolation in fine-tuning efficiency, requiring fewer steps to adapt to the target length (Chen et al., 2023b), though it performs poorly in training-free settings (Zhuo et al., 2024). Advanced strategies such as NTK (bloc97, 2023) and YaRN (Peng et al., 2023) have demonstrated decent training-free performance while being more efficient in fine-tuning scenarios. Further refinements, such as optimizing RoPE frequencies (Ding et al., 2024) or modifying RoPE's extrapolation behavior (Hu et al., 2024), have shown additional improvements in language modeling. Our work provides new insights into the impact of RoPE in video diffusion transformers, introducing a length extrapolation strategy tailored for video generation. Unlike previous approaches, our proposed RIFLEx requires training only on the original pre-trained sequence length while also demonstrating strong potential in training-free settings.

**Text-to-video diffusion models.** Drawing upon the progress made in image generation, a burgeoning body of research has emerged, focusing on the utilization of diffusion models for video generation (Kong et al., 2024; Yang et al., 2024; Ho et al., 2022; Polyak et al., 2024; Brooks et al., 2024; Zhou et al., 2024; Team, 2024b; Zheng et al., 2024; Blattmann et al., 2023; Lin et al., 2024; Xing et al., 2023; Chen et al., 2023a; 2024; He et al., 2022; Zhao et al., 2023; 2022). By combining spatial and temporal attention, VDM (He et al., 2022) introduces a space-time factorized UNet for video synthesis, marking an early contribution to the field. Later, Make-A-Video extends the 2D-UNet with temporal modules (Singer et al., 2022), exploring the integration of prior knowledge from text-to-image diffusion models into video diffusion techniques. More recently, a surge of video diffusion models leveraging the expressive power of transformers has emerged (Lin et al., 2024; Zheng et al., 2024; Kong et al., 2024; Yang et al., 2024; Bao et al., 2024; Brooks et al., 2024; Team, 2024b). These diffusion transformer-based models have demonstrated remarkable performance. Our approach builds on these advancements by applying them to pre-trained video diffusion transformers, further enhancing their capabilities. Moreover, recent developments have also seen the emergence of video diffusion models that leverage efficient attention mechanisms to accelerate their performance (Zhang et al., 2024b;a; 2025). Our approach is also applicable to these models, further extending their capabilities.

**Autoregressive video generation models.** Unlike diffusion models, autoregressive video generation models typically quantize videos into discrete tokens and generate video content through next-token prediction in an autoregressive manner. Previous works have demonstrated great performance in such models (Wu et al., 2021; Yan et al., 2021; Hong et al., 2022; Wu et al., 2022; Kondratyuk et al., 2023; Wu et al., 2024; Sun et al., 2024; Wang et al., 2024b). For example, NÜWA (Wu et al., 2022) employs VQ-GAN for tokenization and generates videos using a 3D transformer encoder-decoder framework. More recently, VideoPoet (Kondratyuk et al., 2023) tokenizes images and videos with a MAGVIT-v2 encoder and autoregressively generates videos using a decoder-only transformer based on a pretrained large language model. While autoregressive video models can theoretically extend sequences indefinitely through next-token prediction (Wang et al., 2024c; Liang et al., 2022; Ge et al., 2022), recent studies reveal their tendency to degenerate into repetitive content generation (Kondratyuk et al., 2023; Ge et al., 2022). In this work, we present a principled approach to video length extrapolation that effectively generates novel temporal content in diffusion-based frameworks. Although our method is developed for video diffusion transformers, the underlying mechanism governing position embedding periodicity may also offer insights for addressing repetition challenges in autoregressive video generation.

**Long video with diffusion models.** Recent studies have explored long video generation with diffusion models from various angles (Lu et al., 2024a; Wang et al., 2023; 2024a;c; Lin et al., 2023; Li et al., 2024; Qiu et al., 2023; NVIDIA et al., 2025). For instance, Kim et al. (2024); Chen et al. (2025) propose diffusion sampling schemes that employ a queue of video frames with varying noise levels, progressively decoding new frames. Yan et al. (2024) introduce a cross-attention module to enhance the semantic fidelity and richness of long videos. Yin et al. (2024) distill a chunk-wise, few-step auto-regressive video diffusion transformer from a bidirectional teacher model, enabling efficient long video generation. In this work, we address long video generation with diffusion transformers through the lens of position encoding—a fundamental component for capturing sequential structure in video data. We propose a minimal yet general and effective strategy that requires no training on long video data.

# B. Additional Results of RIFLEx

In this section, we present additional demos for temporal extrapolation in Fig. 6, spatial extrapolation in Fig. 7, and both extrapolations in Fig. 8.

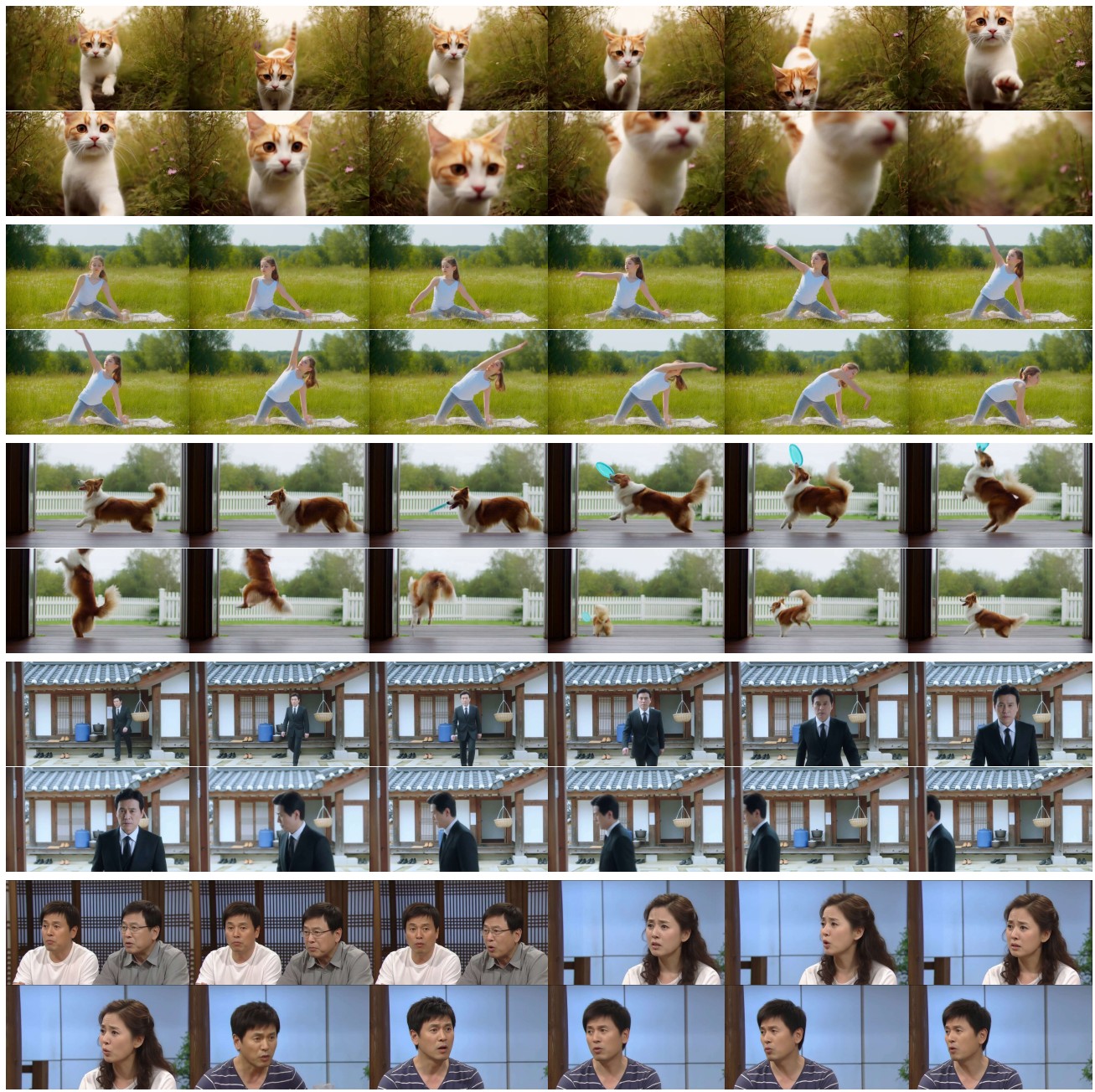

Figure 6. **More results of** $2\times$ **temporal extrapolation from** $129$ **to** $261$ **frames.**

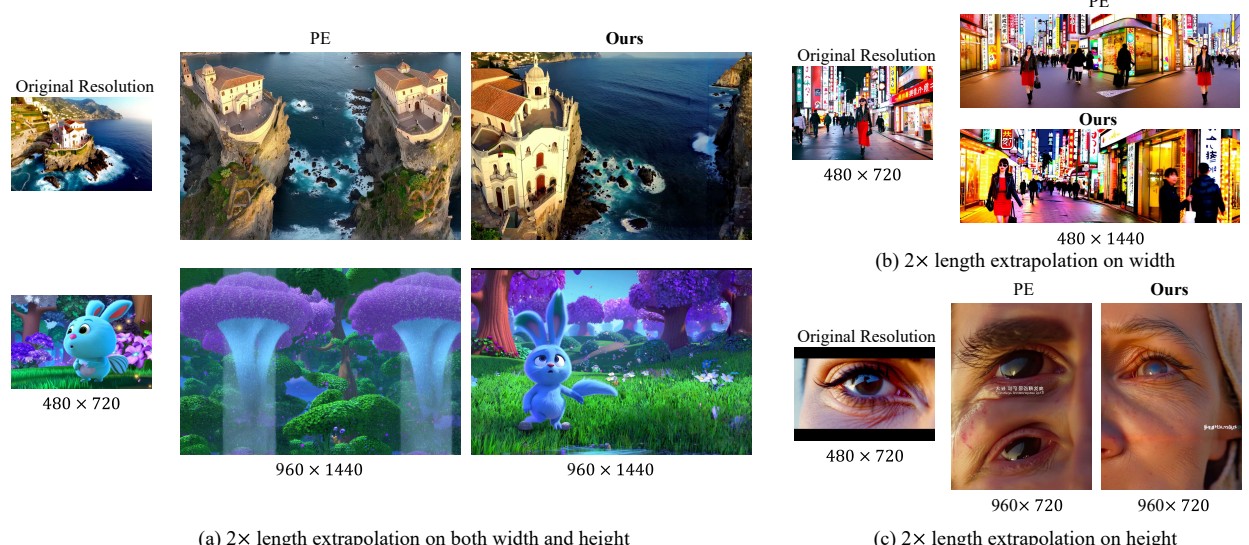

(a) 2× length extrapolation on both width and height

(b) 2× length extrapolation on width

(c) 2× length extrapolation on height

*Figure 7.* **Visualization results of spatial resolution extrapolation method in image generation.** Our method outperforms the extrapolation by generating new content with better visual quality.

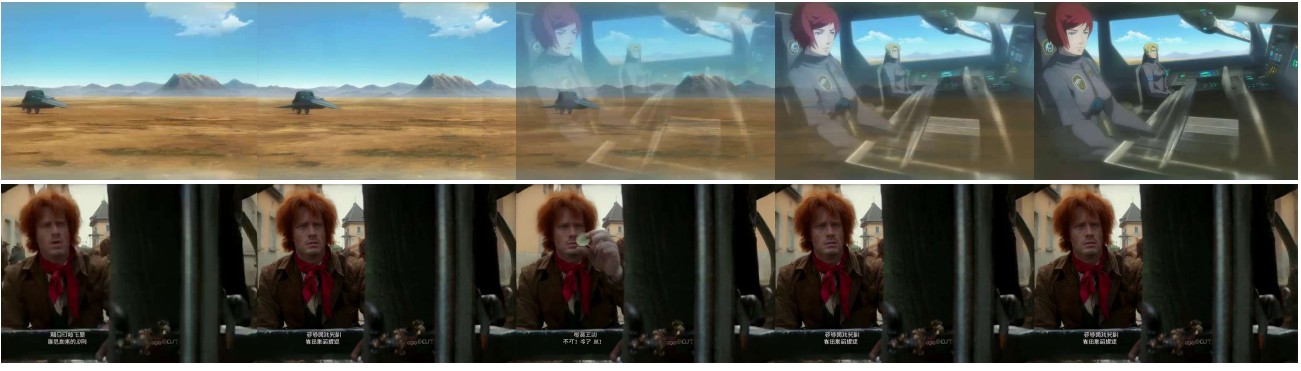

*Figure 8.* **More results of** 2× **temporal and spatial extrapolation**, extending video dimensions from 480 × 720 × 49 to 960 × 1440 × 97.

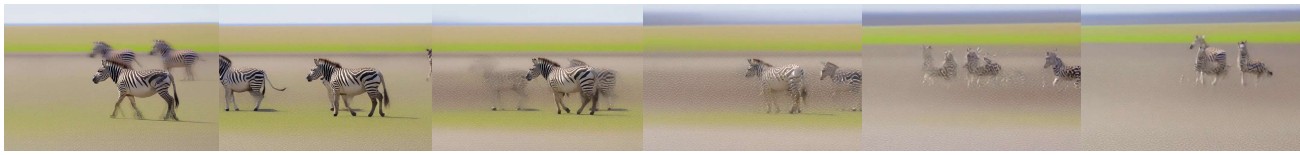

*Figure 9.* **Results of** 4× **temporal extrapolation from** 49 **to** 193 **frames.**

*Table 2.* **Code Links and Licenses.**

| Method | Link | License |
|---|---|---|
| HunyuanVideo (Kong et al., 2024) | https://github.com/Tencent/HunyuanVideo | Tencent Hunyuan Community License |
| FastVideo (Team, 2024a) | https://github.com/hao-ai-lab/FastVideo | Apache License |
| CogVideoX (Yang et al., 2024) | https://github.com/THUDM/CogVideo | Apache License |
| Lumina-T2X (Zhuo et al., 2024) | https://github.com/Alpha-VLLM/Lumina-T2X | MIT License |

*Figure 10.* **Visualization of other existing methods for $2\times$ extrapolation in video and image generation.** YaRN leads to slower motion. While TASR can successfully perform resolution extrapolation, it simultaneously causes slower motion and temporal repetition in video generation.

## C. More Results of Failure Patterns of Existing Methods

As shown in Fig. 10, we present the results of other existing methods for $2\times$ extrapolation in video and image generation. Specifically, YaRN results in slower motion, using parameters $\alpha = 1$ and $\beta = 32$ as set in previous studies (Lu et al., 2024b; Peng et al., 2023). TASR utilizes PI at larger timesteps and employing NTK at smaller timesteps. Consequently, it combines the characteristics of both PI and NTK, which leads to slower motion and temporal repetition in video generation.

## D. Experimental Setup.

**Used code and license.** All used codes in this paper and its license are listed in Tab. 2.

**Implementation details.** For spatial extrapolation, following Algorithm 1, we identify the intrinsic frequency components whose periods closely match the repeating patterns observed in the height and width pixels, then adjust them to ensure unique encoding. For both spatial and temporal extrapolation, we simultaneously adjust the intrinsic frequency components for the time, width, and height dimensions. The training-free setting shares the same intrinsic frequency values as those in Tab. 3.

**Evaluation metrics.** For the NoRepeat Score, we identify the frame around $N_k$ with the minimum $L_2$ distance to the first frame, marking it as the start of the possible repeated sequence. We then calculate the $L_2$ distance between each frame in the possible repeated sequence and the corresponding frame at the beginning of the video. If the average distance across frames exceeds a threshold, the video has a higher probability of being non-repetitive. We then calculate the proportion of videos with a higher probability of being non-repetitive. Empirically, we find that a threshold of 100 aligns better with human perception, so we set it to 100. For the human evaluation of the training-free setting, considering that several methods may share similar quality (e.g., slow motion with poor visual quality), we allow for ties. However, for the fine-tuning setting, ties are not permitted.

*Table 3.* **Fine-tuning settings for all experiments**. Both. denotes spatial and temporal extrapolation simultaneously. $b_k^{t'}$, $b_k^{h'}$, and $b_k^{w'}$ represent the base frequency for the intrinsic frequency in the time, height, and width dimensions, respectively. By adjusting these variables, we can modify the corresponding $\theta_k^{t'}$, $\theta_k^{h'}$, and $\theta_k^{w'}$ values accordingly (refer to Section 2.2 for details).

| Config | $2\times$ **Temporal** | $2\times$ **Temporal** | $3\times$ **Temporal** | $2\times$ **Spatial** | $2\times$ **Both.** |
|---|---|---|---|---|---|
| Base model | CogVideoX-5B | HunyuanVideo | CogVideoX-5B | CogVideoX-5B | CogVideox-5B |
| Training iterations | 2500 | 1000 | 5000 | 2000 | 10000 |
| $b_k^{t'}$ | 1e5 | 560 | 1e6 | - | 1e5 |
| $b_k^{h'}$ | - | - | - | 1e6 | 1e6 |
| $b_k^{w'}$ | - | - | - | 5e4 | 5e4 |
| Data size | $480 \times 720 \times 49$ | $544 \times 960 \times 129$ | $480 \times 720 \times 49$ | $480 \times 720 \times 1$ | $480\times720\times49$ |
| Batch size | 8 | 8 | 8 | 64 | 8 |
| GPU | 8 A100-80G | 24 A100-80G | 8 A100-80G | 8 A100-80G | 8 A100-80G |

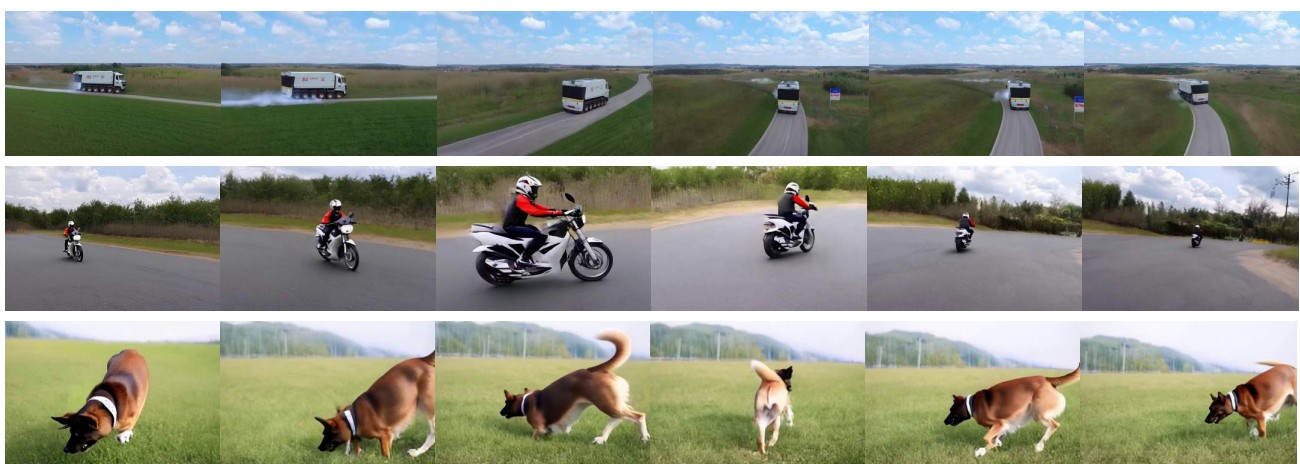

*Figure 11.* **The results of adjusting all frequency components lower than the intrinsic frequency.** See detailed analysis in Appendix E.

## E. Details about RIFLEx

**Robustness of the intrinsic frequency $k$.** Empirically, we collected 20 videos and found that, although the first observed repetition frame may vary across videos within a certain range, the identified intrinsic frequencies remain consistent. For example, in HunyuanVideo, even though the first observed repetition frame range from 178 to 200, the closest intrinsic frequency is always $k = 4$, where $N_k = 200$.

**Adjust all frequency components lower than the intrinsic frequency.** In our preliminary experiments, we slow down all frequency components lower than the intrinsic frequency by increasing the base frequency $b$ for $j \geq k$, where $b$ is chosen to satisfy the non-repetition condition Eqn. (8) for intrinsic frequency $k$. As shown in Fig. 11, this approach effectively addresses the repetition issue while maintaining visual quality. It is important to note that, despite this choice, our RIFLEx, which reduces the intrinsic frequency, provides the minimal solution.

**Ablations for other frequencies.** As shown in Fig. 12, reducing the higher-frequency components slows down the video. Based on the analysis in Section 3.2, this may be because these components are crucial for capturing fast motion. Reducing their frequencies leads to a slower rate of change in the positional encoding, which weakens the model's ability to generate rapid movements.

On the other hand, reducing the lower frequencies has a negligible impact. This is likely because, for these frequencies, the encoding functions change very little across the training length, from $p = 1$ to $p = L$. Therefore, these frequencies may be less sensitive to positional encoding, and altering them results in minimal effect.

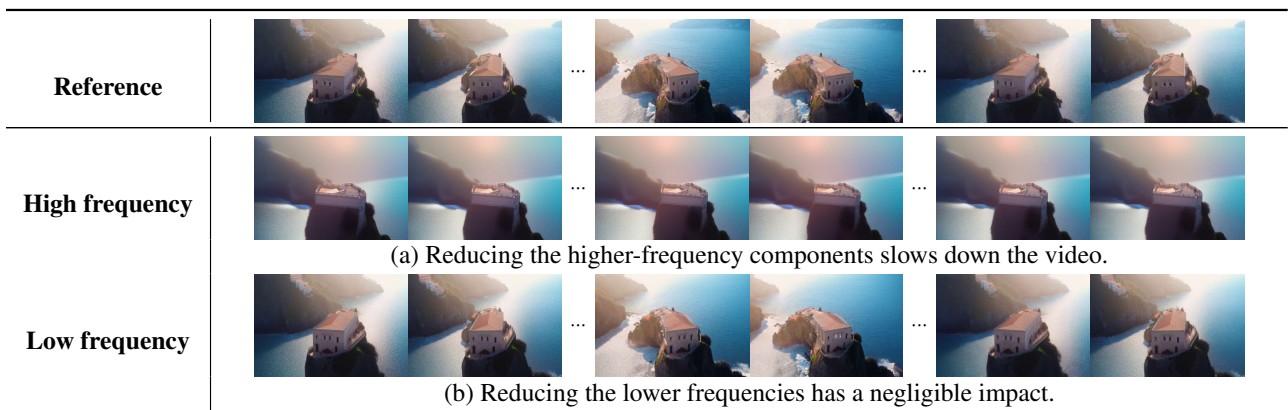

(a) Reducing the higher-frequency components slows down the video.

(b) Reducing the lower frequencies has a negligible impact.

*Figure 12.* **Ablations for reducing other frequencies.** Reference refers to the results of PE, where no frequencies are reduced, serving as the baseline.

## F. More Results about Comparisons

In this section, we show the visualization comparisons of CogVideoX-5B. As shown in Fig. 13, PI and YaRN suffer from slow motion, while PE and NTK experience repetition issues. TASR suffers from both slow motion and repetition. By effectively addressing both challenges, our method significantly enhances motion quality.

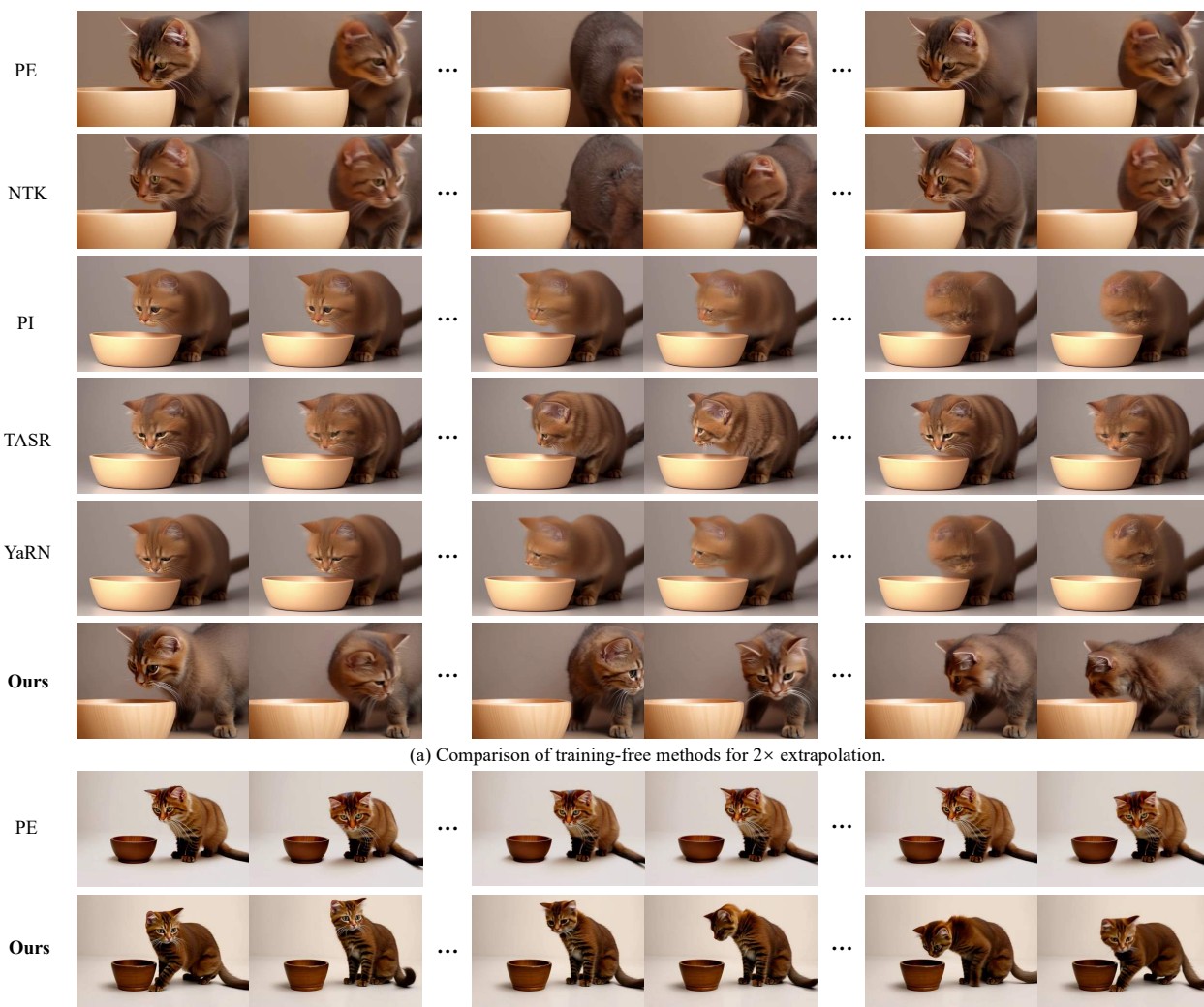

(a) Comparison of training-free methods for 2× extrapolation.

(b) Comparison of fine-tuning methods for 2× extrapolation.

*Figure 13.* **Visualization results of length extrapolation based on CogVideoX-5B.** We achieve better video quality by effectively addressing issues of slow motion and repetition.

