# OpenReview forum: "RIFLEx: A Free Lunch for Length Extrapolation in Video Diffusion Transformers"
_ICML.cc/2025/Conference — ICML 2025 poster_

### Official Review · Reviewer_DWAu · 2025-02-17

**Overall Recommendation:** 4

**Summary:**

This paper propose a training-free method for video extrapolation. It argues that existing extrapolation strategies, originally developed for text and image generation, fail on videos because of temporal repetition and slow motion. It analyzes the frequency components in positional encoding, isolating individual frequency components by zeroing out others and fine-tuning the target video model. It finds that high frequencies capture short-term dependencies and induce temporal repetition, while low frequencies encode long-term dependencies but lead to motion deceleration. Furthermore, this paper identifies a consistent intrinsic frequency component across different videos from the same model, which primarily  dictates repetition patterns among all components during extrapolation. Based on the observation, this paper proposes to lowers the intrinsic frequency to ensure it remains within a single cycle after extrapolation. In addition, this technique can also be applied to spatial extrapolation.

Experiments are on state-of-the-art video diffusion transformers, including CogVideoX-5B and HunyuanVideo for 2x extrapolation. Besides the training-free method, the paper also explores the possibility of fine-tuning, which improves the sample quality and extends to 3x extrapolation.

## update after rebuttal
The rebuttal has addressed my concerns and questions. Since I already gave accept, I will keep the score.

**Claims And Evidence:**

1. Claim: A comprehensive understanding of video length extrapolation.

  Evidence: Qualitative results show that the positional encoding based previous extrapolation methods gives either repeated frames or slow motion. Quantitative results show that there exists a consistent intrinsic frequency component across different videos from the same model.

2. Claim: A training-free extrapolation solution by reducing the intrinsic frequency.

  Evidence: Qualitative results show that reducing the intrinsic frequency indeed help in video extrapolation.

3. Claim: 2x extrapolation in the training-free manner and 3x extrapolation in the fine-tuning manner.

  Evidence: Qualitativer results shows fine-tuning is necessary for 3x extrapolation. The fine-tunning takes 20,000 original length videos and 1/50,000 of the pre-training computation.

**Essential References Not Discussed:**

N/A

**Experimental Designs Or Analyses:**

The experiments look sound.

**Methods And Evaluation Criteria:**

The method is simple yet effective. By only reducing the intrinsic frequency for those video generation models, the extrapolation length is significantly longer. The effectiveness is demonstrated quantitatively by NoRepeat Score and Dynamic Degree, and qualitatively by the supplementary videos.

**Other Comments Or Suggestions:**

N/A

**Other Strengths And Weaknesses:**

N/A

**Questions For Authors:**

If only 20,000 videos are needed, how to select them? Does the selection make any difference?

**Relation To Broader Scientific Literature:**

In a broader sense, I believe this pre-context aware video extrapolation is also related to video prediction. The key of both areas is how to generate the next frames with the context of existing frames.

**Theoretical Claims:**

I checked the frequency analysis in RoPE, which makes sense.

---

> ### Author Rebuttal · Authors · 2025-04-01
>
> We sincerely thank Reviewer DWAu for the recognition of our work. The further questions are addressed as follows.
>
> ### Q1: If only 20,000 videos are needed, how to select them? Does the selection make any difference?
>
> The 20K videos in this paper were randomly sampled without selection. Fine-tuning aims to adapt the model to modified frequencies, which theoretically requires no data bias. To validate this, we add an experiment where we independently train models on two distinct randomly sampled datasets. We then perform three sampling runs with different random seeds and apply a two-sample $t$-test to compare the performance of the two models. As demonstrated in Rebuttal Table A, the statistical analysis reveals no significant performance difference between the models at the $95$% confidence level (all two-sample $t$-test $p$-values > $0.05$, $α=0.05$).
>
> **Rebuttal Table A.** Performance comparison between Model A and Model B, trained on two independent randomly sampled datasets based on the CogVideoX-5B architecture. Evaluation is conducted on a 165-sample subset of VBench with three sampling runs, reporting the mean ± standard deviation. The p-values are derived from a two-sample $t$-test.
> | Metric| Model A / Model B| P Value|
> |--|--|--|
> |NoRepeat Score| 81.21 $\pm$ 7.572 / 81.82 $\pm$ 6.414 | 0.9212|
> |Dynamic Degree| 53.24 $\pm$ 3.493 / 62.03 $\pm$ 7.650 | 0.1444|
> |Imaging Quality| 60.30 $\pm$ 0.8279 / 58.96 $\pm$ 0.8874 | 0.1270|
> |Overall Consistency| 25.26 $\pm$ 0.2974 / 25.21 $\pm$ 0.2053| 0.7994|

---

### Official Review · Reviewer_Cp7v · 2025-03-11

**Overall Recommendation:** 3

**Summary:**

This paper focused on video length extrapolation in Video Diffusion Transformers. The authors provided a comprehensive understanding of video length extrapolation by analyzing the role of frequency components in RoPE. Furthermore, a minimal yet effective method named RIFLEx is proposed to prevent repetition by reducing intrinsic frequency. Experimental results show that RIFLEx achieves high-quality 2× extrapolation on state-of-the-art video diffusion transformers in a training-free manner.

**Claims And Evidence:**

This paper claims that "generating even longer videos with temporal coherence remains a major challenge and existing length extrapolation methods lead to temporal repetition or motion deceleration." and validates this by experimental results.

**Essential References Not Discussed:**

No

**Experimental Designs Or Analyses:**

Yes, a minimal yet effective method named RIFLEx is proposed to prevent repetition by reducing intrinsic frequency.

**Methods And Evaluation Criteria:**

A minimal yet effective method named RIFLEx is proposed to prevent repetition by reducing intrinsic frequency. Qualitative and quantitative evaluations are conducted.

**Other Comments Or Suggestions:**

Please refer to Weaknesses for more details

**Other Strengths And Weaknesses:**

Pros:

1.	A minimal yet effective method named RIFLEx is proposed to prevent repetition by reducing intrinsic frequency.
2.	This paper is well written and easy to follow.
3.	Experimental results verify the effectiveness of the proposed method.

Cons:

1.	The experimental results shown in the supplementary materials display that some cases may still suffering from temporal inconsistency and may lead to the camera to switch in the playing video. Could the authors give some explanation?
2.	Why is the effect not good when the insertion multiple is greater than 3? So how do we choose the appropriate multiples according to different models?

**Questions For Authors:**

Please refer to Weaknesses for more details

**Relation To Broader Scientific Literature:**

This paper focused on video length extrapolation in Video Diffusion Transformers. The authors provided a comprehensive understanding of video length extrapolation by analyzing the role of frequency components in RoPE. Furthermore, a minimal yet effective method named RIFLEx is proposed to prevent repetition by reducing intrinsic frequency. Experimental results show that RIFLEx achieves high-quality 2× extrapolation on state-of-the-art video diffusion transformers in a training-free manner.

**Theoretical Claims:**

No theoretical claims

---

> ### Author Rebuttal · Authors · 2025-04-01
>
> We thank reviewer Cp7v for the valuable comments. We address the concerns as follows.
>
> ### Q1: The experimental results shown in the supplementary materials display that some cases may still suffering from temporal inconsistency and may lead to the camera to switch in the playing video. Could the authors give some explanation?
>
> We appreciate the reviewer's attention to this detail and would like to clarify that this may be a potential misunderstanding. In fact, **multi-scene generation with camera transitions is a desirable and essential capability for video synthesis**. To achieve this, HunyuanVideo is specifically designed to curate a training dataset that includes diverse scene transitions (see Dense Description in Section 3.2 of HunyuanVideo[1]). As demonstrated on HunyuanVideo's project page[2] (e.g., the example in Row 2, Column 1), HunyuanVideo excels at "breaking monotonous cinematography with seamless director-level shot transitions." This capability represents a significant advancement in video synthesis.
>
> **Our method preserves the base model's ability to generate both multi-scene videos (Videos 1-3,9 in the supplementary materials) and single-scene videos (Videos 4-8)**. Importantly, even in multi-scene generation, our method still maintains long-term temporal consistency—for instance, Figure 1 demonstrates consistent identity preservation across two distinct scene transitions. We will clarify this scene transition capability in Section 4.2 of the final version.
>
> [1] Kong, Weijie, et al. "Hunyuanvideo: A systematic framework for large video generative models." arXiv preprint arXiv:2412.03603 (2024).
>
> [2] https://aivideo.hunyuan.tencent.com/.
>
>
>
>
> ### Q2: Why is the effect not good when the insertion multiple is greater than 3? So how do we choose the appropriate multiples according to different models?
>
> #### Q2-1: Why is the effect not good when the insertion multiple is greater than 3?
>
> The limitation of a 3× insertion multiple stems from a fundamental trade-off in positional encoding dynamics: an excessive reduction in frequency leads to a diminished ability to discriminate between sequential positions. Specially, a larger extrapolation factor $s$ leads to smaller frequencies $θ_k$ (see Eqn.(8)).  When $θ_k$ becomes excessively small, the position difference term $∆ = \cos((p + 1)θ_k) − \cos(p\theta_k)$ diminishes to near-zero values, thereby losing positional discriminability. Empirically, we find the threshold occurs at $θ_k′ \le 2\pi/3L$ when $s=3$. Furthermore, **this upper limit is inherent to the positional encoding mechanism and is consistent across existing pretrained video diffusion models** based on our experiments. We will add the above detailed explanation in "Maximum extent of extrapolation" part of the final version.
>
> #### Q2-2: So how do we choose the appropriate multiples according to different models?
>
> We clarify that **we do not select different $s$ for different models**. Instead, the extrapolation length is determined by the user's requirements, provided it remains below the upper limit. For example, in the paper, we demonstrate results for generating videos with multipliers of $2$, $2.3$, and $3$. Based on the analysis in Q2-1, the extrapolation limit is consistent across current models, eliminating the need for specific designs tailored to different models.
>
> We hope you might find the response satisfactory, and we would be delighted to clarify any further concerns you might have.

---

### Official Review · Reviewer_JAfh · 2025-03-14

**Overall Recommendation:** 4

**Summary:**

This paper focuses on a challenging question: How to do the length extrapolation for a trained video diffusion model? After some systematic analyses, they found a metric, named intrinsic frequency that governs the extrapolation property of a video diffusion model. Then, they propose  RIFLEx to reduce the intrinsic frequency when generating longer videos. This method is valid at preserving the motion consistency and preventing the content repetition. The method enables high-quality 2× extrapolation in a completely training-free manner and supports 3× extrapolation with minimal fine-tuning on original-length videos.

## update after rebuttal
I appreciate that the authors take time to provide additional explanations. My concerns are fully addressed. I will keep my original score and support its acceptance.

**Claims And Evidence:**

Yes.

**Essential References Not Discussed:**

Related work on repetition issues in autoregressive video generation models

**Experimental Designs Or Analyses:**

Yes, the experimental designs are sound and comprehensive.

**Methods And Evaluation Criteria:**

Yes

**Other Comments Or Suggestions:**

See Other Strengths And Weaknesses

**Other Strengths And Weaknesses:**

Strengths:
1. The solution is simple and effect.
2. The overall writing is good and easy to follow.
Weaknesses:
1. Limited exploration of extrapolation factors beyond 3×
2. The identification method for intrinsic frequency relies on visual inspection rather than a theoretical analysis.

**Questions For Authors:**

See Other Strengths And Weaknesses

**Relation To Broader Scientific Literature:**

This work effectively builds upon some previous research directions, like position embedding in diffusion model.

**Theoretical Claims:**

N/A - The paper doesn't present formal proofs for theoretical claims.

---

> ### Author Rebuttal · Authors · 2025-04-01
>
> We appreciate Reviewer JAfh for the acknowledgement of our contributions.
>
> ### Q1: Missing related work on repetition issues in autoregressive video generation models
>
>
> Unlike diffusion models, autoregressive video generation models typically quantize videos into discrete tokens and generate video content through next-token prediction in an autoregressive manner. Previous works have demonstrated great performance in such models [1–8]. For example, NÜWA [4] employs VQ-GAN for tokenization and generates videos using a 3D transformer encoder-decoder framework. More recently, VideoPoet [5] tokenizes images and videos with a MAGVIT-v2 encoder and autoregressively generates videos using a decoder-only transformer based on a pretrained large language model.
>
> While autoregressive video models can theoretically extend sequences indefinitely through next-token prediction [9-11], recent studies reveal their tendency to degenerate into repetitive content generation[5,11]. In this work, we present a principled approach to video length extrapolation that effectively generates novel temporal content in diffusion-based frameworks. Although our method is developed for video diffusion transformers, the underlying mechanism governing position embedding periodicity may also offer insights for addressing repetition challenges in autoregressive video generation.
>
> Thank you for highlighting this important direction. We will incorporate the above discussion in related work section in the final verion. We would appreciate any additional references the reviewer could suggest that we may have missed.
>
>
> [1] GODIVA: Generating Open-Domain Videos from Natural Descriptions
>
> [2] VideoGPT: Video Generation using VQ-VAE and Transformers
>
> [3] CogVideo: Large-scale Pretraining for Text-to-Video Generation via Transformers
>
> [4] NÜWA: Visual Synthesis Pre-training for Neural Visual World Creation
>
> [5] VideoPoet: A Large Language Model for Zero-Shot Video Generation
>
> [6] VILA-U: a Unified Foundation Model Integrating Visual Understanding and Generation
>
> [7] Generative Multimodal Models are In-Context Learners
>
> [8] Emu3: Next-Token Prediction is All You Need
>
> [9] Loong: Generating Minute-level Long Videos with Autoregressive Language Models
>
> [10] NUWA-Infinity: Autoregressive over Autoregressive Generation for Infinite Visual Synthesis
>
> [11] Long Video Generation with Time-Agnostic VQGAN and Time-Sensitive Transformer
>
>
>
> ### Q2: Limited exploration of extrapolation factors beyond 3×
>
> Thank you for the suggestions. In this work, we primarily focus on achieving length extrapolation at minimal cost based on a pre-trained video diffusion models. As discussed in the main text, the 3× limitation stems from diminished ability to discriminate sequential positions due to excessive frequency reduction. To further extend beyond the 3× extrapolation, it is promising to investigate the mechanism of positional encoding during training, specifically tailored for extrapolation. We believe our findings colud provide valuable insights for this direction, and we will include a more detailed discussion in Section 5.
>
>
>
> ### Q3: The identification method for intrinsic frequency relies on visual inspection rather than a theoretical analysis.
>
> Thank you for the insightful comment. In our current work, we primarily adopt an empirical approach—visual inspection—for intrinsic frequency identification when adapting the pre-trained video diffusion transformer. While this approach is effective for adaptation, we agree that establishing a theoretical foundation for intrinsic frequency identification is crucial. Achieving this would require fundamental research into how intrinsic frequencies emerge during the pre-training process, potentially analysis from a training-from-scratch perspective. We sincerely thank the reviewer for highlighting this direction, and we will address this in our future work. We will add the above points to the discussion section.

---

### Official Review · Reviewer_EXby · 2025-03-14

**Overall Recommendation:** 4

**Summary:**

This work solves the problem of repetitiveness in long video generation from a new perspective. This work first analyzes and experiments the frequency component of the video position encoding ROPE, and concludes that the period of the frequency component directly affects the periodicity of certain characteristics of the generated video, and the frequency component closer to the video repetition period has the greatest impact. Therefore, based on this idea, this work defines an intrinsic frequency component, which has a period close to the video repetition period. Then, directly reducing this frequency component can alleviate the problem of video repetition. This method can be used in the latest and most advanced video generation models, such as CogVideoX and HuanyuanVideo, which have their corresponding intrinsic frequency components. Therefore, for video extrapolation, this method can achieve two training-free efficient fine-tuning methods:

1) For training-free manner, the "free lunch" is to directly find the corresponding intrinsic frequency component, and then directly reduce this frequency component according to the extrapolation factor.

2) For a larger extrapolation factor, a very small number of training samples can be used to fine-tune the video generation model to adapt to the reduced frequency component.

The author conducted extensive experiments on this method in CogVideoX and HuanyuanVideo, and found that it outperformed other existing video extrapolation methods in both quantitative evaluation and quality assessment. This work alleviates the problem of video extrapolation from a completely new perspective at the lowest cost, which is of great significance to the field of video generation.

## update after rebuttal

I would like to thank the authors for their responses in the rebuttal, which addressed my concerns. I think this is a good paper and should be accepted.

**Claims And Evidence:**

- High frequencies capture rapid movements and short-term dependencies, inducing temporal repetition, while low frequencies encode long-term dependencies with slow motion. This conclusion is based on the calculation of the period of the frequency component and the repetition period of the corresponding video extrapolation. The basis here is supported by both theoretical deduction and experimental demonstration.
- Since the intrinsic frequency component directly affects the repetition period of the video, reducing the intrinsic frequency component lengthens the repetition period of the video, so that video extrapolation can be extended to longer videos.

**Essential References Not Discussed:**

NA

**Experimental Designs Or Analyses:**

- In the quantitative evaluation of the experimental part, in the training-free evaluation of HunyuanVideo, the proposed method is slightly worse than the best method in automatic metrics. The author can analyze the specific reasons in detail.
- From a theoretical point of view, the biggest difference between the method proposed by the author and previous methods such as PI, NTK or YaRN is that the method in this paper focuses on the most important intrinsic frequency component rather than a group of components. Is this correct? What is the importance of this difference?

**Methods And Evaluation Criteria:**

- This work proposes a novel "free lunch" for video extrapolation. This method is very simple and clear, and is a perfect solution from derivation to practical application. This method has been convincingly verified on the latest video generation model.
- For longer video extrapolation, the method proposed in this work requires simple fine-tuning, which may be due to the gap between theoretical derivation and practical application. However, experiments show that only a small amount of fine-tuning is required, and the method can be adapted to longer video extrapolation tasks.
- This work uses the common evaluation criteria for video generation to evaluate the results of video extrapolation, and has achieved amazing results in both quantitative evaluation and quality evaluation.

**Other Comments Or Suggestions:**

- Line 5 in Algorithm 1 should not refer to Eqn. (8), but to the equation in 306, that is, $\theta'_{k}=\frac{2\pi}{Ls}$.
- The subscripts in the formulas in the article basically refer to frequency components. The author should provide the definition of the subscripts multiple times to prevent people from misunderstanding them as frame subscripts. This is a minor concern of mine.

**Other Strengths And Weaknesses:**

Strengths:

The author has carried out detailed theoretical derivation and experimental verification, showing the correlation between the intrinsic frequency component and the video repetition period. This method is very simple and clear, and is a perfect solution from derivation to practical application.

Weaknesses:
- What is the core difference of this method compared with other methods? Is it to scale only the most critical frequency component, rather than a group of frequency components? The paper can discuss in detail the difference from previous methods.
- What is the reason why the method proposed in the paper will have a gap between theory and practice? Why can fine-tuning alleviate this problem? The author can analyze this problem in detail.
- Why does this set of ideas work in spatial extrapolation?

**Questions For Authors:**

NA

**Relation To Broader Scientific Literature:**

This work alleviates the problem of video extrapolation from a completely new perspective at the lowest cost, which is of great significance to the field of video generation.

**Theoretical Claims:**

I checked the theoretical derivation and proof and found no obvious problems.

---

> ### Author Rebuttal · Authors · 2025-04-01
>
> We sincerely thank Reviewer EXby for the valuable suggestions. We have thoroughly addressed the detailed comments as follows.
>
> ### Q1: Explain the proposed method is slightly worse than the best method in automatic metrics.
>
> We kindly clarify that **only through a comprehensive consideration of multiple metrics can we objectively assess video generation quality, rather than relying solely on a single metric.** For example, while PI and YaRN achieve the highest score in the single NoRepeat Score metric, it performed significantly worse in the Dynamic Degree metric (PI/YaRN vs. our RIFLEx in Table 1), resulting in poor video quality and rendering other metrics meaningless. See Figure 5 for the evidence. To highlight this, we mark severe issues in red for clarity (see lines 330-332). We emphasize that **our method is the only one consistently highlighted in the green zone across all 5 settings in Table 1**.
>
> ### Q2: The core difference between this method and others.
>
>
> Our method differs from prior works in two key aspects:
>
> - We determine **which frequency in RoPE should be modified**—specifically, the intrinsic frequency whose period aligns with the first observed repetition frame in a video (Eqn. (7)). As discussed in the main text (lines 306–311), modifying only this frequency is sufficient: adjusting higher frequencies disrupts fast motion, while altering lower frequencies has negligible impacts.
> - We derive **how to adjust this frequency**—ensuring it remains within a single period after extrapolation (the non-repetition condition, Eqn. (8)).
>
> Existing approaches modify multiple frequency components in RoPE, but **they may target incorrect components**. For instance, methods like YaRN and PI mistakenly adjust high frequencies, which results in slow motion (see lines 295–299). Furthermore, even when some methods include the correct components, **their modifications can be flawed**. For example, the adjustment of intrinsic frequency in NTK does not satisfy our non-repetition condition, resulting in repetition issues (see lines 326–329).
>
>
> In summary, our work establishes principled guidelines for position embedding design in length extrapolation. We will make it more clear in the "Principled Explanation for Existing Methods" section (Section 3.4). We hope this response clearly articulates our contributions and would be very happy to clarify further concerns (if any).
>
> ### Q3: Explain the gap between theory and practice and why fine-tuning can alleviate this.
>
> We understand that the gap you are referring to is that fine-tuning the model yields better results than not fine-tuning. This arises from a training-testing mismatch, where the position embeddings used during inference slightly differ from those in training due to modified frequencies. While this discrepancy does not undermine the conclusion about our non-repetition condition, it may affect visual quality since the model lacks explicit training on these specific position embeddings. Fine-tuning helps bridge this gap by adapting the model to these variations, thereby improving visual quality. We will incorporate the above explanation of the training-testing mismatch into Section 3.3, where we discuss whether fine-tuning is a necessary.
>
> ### Q4: Why does this set of ideas work in spatial extrapolation?
>
> This is because video diffusion transformers typically apply 1D RoPE independently to both spatial and temporal dimensions (see Section 2.2, "RoPE with Multiple Axes"). **This shared mechanism results in similar challenges during extrapolation for both dimensions**. As illustrated in Figure 2:
> - Spatial Repetition ↔ Temporal Repetition: Both phenomena occur when intrinsic frequency components exceed a single period after extrapolation.
> - Blurred Details ↔ Slower Motion: These effects arise from interpolating high-frequency components, which are essential for spatial details in spatial domain and fast motion in the temporal domain.
>
> Therefore, our method can be extended to spatial extrapolation, providing a unified framework for extrapolation in diffusion transformers. We will add above explanation in "Extension to other extrapolation types" part of the final version.
>
> ### Q5: Line 5 in Algorithm 1 should refer to $\theta_k' =\frac{2\pi}{Ls}$.
>
> We will correct Line 5 of Algorithm 1 to use $\theta_k' =\frac{2\pi}{Ls}$ in the final version. Thank you for the careful review.
>
> ### Q6: Provide the definition of the subscripts multiple times.
> As suggested, we will explicitly define the subscript $j$ as indexing frequency components of RoPE multiple times in Section 3.1 to prevent confusion with frame indices in the final version.
>
> Finally, we sincerely appreciate the reviewer for the constructive suggestions, which help to further improve the quality of our work. We hope you may find the response satisfactory. Please let us know if you have any further feedback.

---

### Decision · Program_Chairs · 2025-05-01

**Decision:**

Accept (poster)

**Comment:**

All 4 reviewers recommended acceptance (1 weak accept, 3 accepts). After reading the paper and reviews, I agree with the reviewers and recommend acceptance.